# RNA-Seq and 16S rRNA Reveals That Tian–Dong–Tang–Gan Powder Alleviates Environmental Stress-Induced Decline in Immune and Antioxidant Function and Gut Microbiota Dysbiosis in *Litopenaeus vannami*

**DOI:** 10.3390/antiox12061262

**Published:** 2023-06-12

**Authors:** Xiao-Dong Xie, Ying Zhou, Yu-Bo Sun, Shou-Li Yi, Yi Zhao, Qi Chen, Ying-Hong Xie, Mi-Xia Cao, Mei-Ling Yu, Ying-Yi Wei, Ling Zhang, Ting-Jun Hu

**Affiliations:** 1College of Animal Science and Technology, Guangxi University, Nanning 530005, China; 1718304007@st.gxu.edu.cn (X.-D.X.); zhouying@st.gxu.edu.cn (Y.Z.); yubosun@st.gxu.edu.cn (Y.-B.S.); yishouli@st.gxu.edu.cn (S.-L.Y.); zhaoyi@st.gxu.edu.cn (Y.Z.); chenqi@st.gxu.edu.cn (Q.C.); ying-hongxie@st.gxu.edu.cn (Y.-H.X.); yumeiling@gxu.edu.cn (M.-L.Y.); weiyingyi@gxu.edu.cn (Y.-Y.W.); 2College of Animal Science, Anhui Science and Technology University, Chuzhou 233100, China; caomixia66668888@st.gxu.edu.cn; 3Guangxi Scientific Research Center of Traditional Chinese Medicine, Guangxi University of Chinese Medicine, Nanning 530200, China

**Keywords:** *Asparagus cochinchinensis* (Lour.) Merr., *Panax notoginseng* (Burkill) F.H. Chen ex C.H., *Litopenaeus vannamei*, antioxidant function, gut microbiota

## Abstract

Ammonia stress and nitrite stress can induce immune depression and oxidative stress in *Litopenaeus vannami* (*L. vannamei*). Earlier reports showed that *L. vannamei* immunity, resistance to ammonia stress, and resistance to nitrite stress improved after Tian–Dong–Tang–Gan Powder (TDTGP) treatment, but the mechanism is not clear. In this study, three thousand *L. vannamei* were fed different doses of TDTGP for 35 days and then subjected to ammonia and nitrite stress treatments for 72 h. Transcriptome and 16-Seq ribosomal RNA gene sequencing (16S rRNA-seq) were used to analyze hepatopancreas gene expression and changes in gut microbiota abundance in each group. The results showed that after TDTGP treatment, hepatopancreas mRNA expression levels of immunity- and antioxidant-related genes were increased, the abundance of *Vibrionaceae* in the gut microbiota was decreased, and the abundance of *Rhodobacteraceae* and *Flavobacteriaceae* was increased. In addition, after TDTGP treatment, the effects of ammonia and nitrite stress on the mRNA expression of Pu, cat-4, PPAF2, HO, Hsp90b1, etc. were reduced and the disruption of the gut microbiota was alleviated. In short, TDTGP can regulate the immunity and antioxidant of *L. vannamei* by increasing the expression levels of immunity- and antioxidant-related genes and regulating the abundance of *Rhodobacteraceae* and *Flavobacteriaceae* in the gut microbiota.

## 1. Introduction

*Litopenaeus vannamei (L. vannamei)* is currently one of the three most farmed shrimp species in the world. *L. vannamei* are predominantly innately immune, relying on humoral and cellular immune responses to detect and eliminate invading microorganisms [1]. The immune and antioxidant function and gut microbiota of *L. vannamei* are easily affected by changes in the farming environment. For example, a decrease in antioxidant capacity and the immune capacity, as well as gut microbiota disorders, can be induced in *L. vannamei* by drastic changes in salinity and water temperature, and high concentrations of ammonia or nitrite in the water column [2,3,4,5].

Several studies have shown that the gut microbiota plays a key role in host immune system maturation and ongoing training [6]. For example, gut microbiota prevented pathogen overgrowth [7], regulated gut endocrine function [8], and provided energy sources for bioproduction. In addition, host metabolism and the function of peripheral tissues were influenced by the secretory products of the gut microbiota [9]. Recently, the relationship between invertebrate gut microbiota and their growth or health has received much attention [10,11,12].

Plant extracts have been shown to regulate host immunity, antioxidant, and gut microbiota composition in several studies. For instance, growth performance, immunity, and survival were improved in *L. vannamei* infested with White Spot Syndrome Virus (WSSV) after using oral *Argemone mexicana* extract [13]. Dietary Xiao-Chaihu Decoction [14] improved immune function and antioxidant capacity in *L. vannamei*, and significantly reduced abundances of *Vibrio* and increased abundances of *Ruegeria* in the gut.

Tian–Dong–Tang–Gan Powder (TDTGP) consists mainly of the polysaccharide from *Asparagus cochinchinensis* (Lour.) Merr. and the total saponin from *Panax notoginseng* (Burkill) F.H. Chen ex C.H. in a certain ratio. Previous reports have shown that the immunity, anti-ammonia-stress, and anti-nitrite-stress abilities of *L. vannamei* were improved by oral TDTGP [15,16]. Nevertheless, the mechanism of the antioxidant effect, the immunomodulation effect, and the effect of gut microbiota in *L. vannamei* with oral TDTGP are not understood. The aim of this study was to analyze the effects of oral TDTGP on the regulation of antioxidant- and immunity-related genes and the effects of gut microbiota in *L. vannamei* using RNA-Seq and 16S rRNA-seq. The results may provide a theoretical basis for the clinical application of TDTGP.

## 2. Materials and Methods

### 2.1. Experimental Herbs

TDTGP is composed of polysaccharides of *Asparagus cochinchinensis* (Lour.) Merr. (84.08% total polysaccharide purity, 3.1% moisture) and total saponins of *Panax notoginseng* (Burkill) F.H. Chen ex C.H. (mainly composed of 7.20% notoginsenoside R1, 29.39% ginsenoside Rg1, 4.29% ginsenoside Re, 28.38% ginsenoside Rb1, and 6.77% ginsenoside RD for a total of 76.03%), which were mixed with silica (Zhengzhou Yuanze Chemical Co., Zhengzhou, China) in a ratio of 5:1:2. The polysaccharides of *Asparagus cochinchinensis* (Lour.) Merr. and the total saponins of *Panax notoginseng* (Burkill) F.H. Chen ex C.H. were extracted in the same way as in a previous study [16].

Finally, the total polysaccharide content was 56.90%, the total saponin content was 8.93% (0.901% notoginsenoside R1, 3.413% ginsenoside Rg1, 0.481% ginsenoside Re, 3.354% ginsenoside Rb1, and 0.779% ginsenoside RD), and the content of silica and other substances was 31.17% per 1 g TDTGP.

### 2.2. Experimental Diets

The experimental diets were prepared according to Table 1 [16]. They were separately supplemented with TDTGP to replace cellulose in each group (Table 1). Dry ingredients were finely ground and mixed with fish oil. Purified water was added to form a dough, which was extruded with a miner, matured for 5 h at 70 °C, air dried, and sieved into pellets, which were stored at −20 °C.

### 2.3. Stress Experiment

*L. vannamei* were collected from the National Guangxi Specific Pathogen Free *Litopenaeus Vannamei* Breeding Farm, China and randomly divided into 5 groups (3 replicates per group, 200 shrimps per replicate) of 3000 shrimp (length 4.22 ± 0.22 cm and weight 0.79 ± 0.23 g), including a blank control group, stress control group, TDTGP-2 group, TDTGP-4 group, and TDTGP-8 group (n = 200 × 3 = 600). Each group was individually reared in a 2.6 m^3^ concrete pond with seawater salinity of 28 ± 2.0‰, temperature of 28 ± 3.0 °C, pH of 8.0 ± 0.5, and dissolved oxygen of 5 mg·L^−1^ or more. Water was changed 30% daily for 35 days during the feeding experiment. The above diet was fed to each treatment group four times a day.

An ammonia stress test and a nitrite stress test were performed for 72 h at the end of the feeding experiment. Briefly, 30 plastic tanks of 240 L capacity were divided into 10 groups, including the blank control group (C), the ammonia stress control group (A), the nitrite stress control group (N), the TDTGP-4 group (T4), the TDTGP-2 + ammonia stress group (AT2), the TDTGP-4 + ammonia stress group (AT4), the TDTGP-8 + ammonia stress group (AT8), the TDTGP-2 + nitrite stress group (NT2), the TDTGP-4 + nitrite stress group (NT4), and the TDTGP-8 + nitrite stress group (NT8). There were 3 replicates group in each group, and each replicate group contained 30 shrimp.

A quantity of 100 L of seawater (salinity 28 ± 2.0‰) was added to the plastic tanks. In groups A, AT2, AT4, and AT8, NH_4_Cl was used for ammonia stress modeling, adjusted to 46 mg·L^−1^ (molecular concentration of ammonia 15.48 ± 2.04 mg·L^−1^). The concentrations of NaNO_2_ for the N, NT2, NT4, and NT8 groups were adjusted to 20 mg·L^−1^ (molecular concentration of nitrite 2.59 ± 0.38 mg·L^−1^). Ammonia and nitrite concentrations were measured and adjusted every 24 h.

### 2.4. Sample Collection and Processing Methods

At 72 h of the ammonia stress and nitrite stress tests, 14 shrimp were randomly selected from each replicate of per group to collect hemolymph (a total of 42 shrimps per group). At the end of the stress test, three hepatopancreas were aseptically and randomly collected from each replicate group of groups C, A, N, T4, AT4, and NT4 in a single lyophilization tube as one sample (three hepatopancreas samples were collected from each group). In addition, all shrimp guts from each replicate group were collected and pooled as one sample. The hepatopancreas and gut samples were stored in liquid nitrogen for transcriptome and gut microbiome analysis.

### 2.5. Immunological Parameter Analysis

Total antioxidant capacity (T-AOC), superoxide dismutase (SOD), inducible nitric oxide synthase (i-NOS), and acid phosphatase (ACP) were measured using a kit supplied by the Nanjing Jian Cheng Bioengineering Institute, Nanjing, China. The method of Ashida (1983) was used to detect phenoloxidase (PO) activity in hemolymph [16,17,18].

### 2.6. Hepatopancreas Transcriptome Analysis

Total RNA was extracted from the hepatopancreas of the C, A, N, T4, AT4, and NT4 groups using the RNAiso Plus (TakaRa #9108, Takara Biomedical Technology (Beijing) Co., Ltd., Beijing, China). An Agilent 2100 Bioanalyzer (Agilent Technologies, Palo Alto, CA, USA) was used to assess the quality of RNA in each group. A cDNA library was prepared using the NEBNext Ultra RNA Library Prep Kit for Illumina (NEB #7530, New England Biolabs, Ipswich, MA, USA). Illumina Novaseq 6000 was used to sequence the resulting cDNA library. The RNA differential expression analysis, principal component analysis (PCA), and bioinformatic analysis of gene ontology (GO) enrichment analysis and pathway enrichment analysis were performed using the OmicShare tools at www.omicshare.com/tools (accessed on 7 June 2023). The raw sequencing data were deposited at Genome Sequence Archive, Beijing Institute of Genomics (BIG) Data Center (https://bigd.big.ac.cn/ (accessed on 7 June 2023)), no: CRA006957.

### 2.7. Gut Microbiome Analysis

Shrimp gut microbiome DNA was extracted from C, A, N, T4, AT4, and NT4 groups using a HiPure Stool DNA Kit (Magen, Guangzhou, China). The V3 + V4 region of the ribosomal RNA gene was amplified by PCR. Amplified products were purified, quantified, pooled equimolarly, and paired-end sequenced (PE250) on an Illumina platform (Illumina Novaseq 6000). The Omicsmart online platform (http://www.omicsmart.com (accessed on 7 June 2023)) was used for bioinformatic analysis of the raw data, including operational taxonomic units (OTUs), community composition, and alpha and beta diversity analysis. The data were deposited in BIG Data Center, under accession no: CRA006959.

### 2.8. Quantitative Real-Time PCR Validation

Seven differentially expressed genes were randomly selected from the sequencing results in groups C, A, N, T4, AT4, and NT4 for qPCR validation, and consistent expression patterns indicated high confidence in the transcriptome data (Appendix A). The abm kits (All-In-One 5 X RT MasterMix Cat#G592) were used to transcribe RNA into using cDNA. Primers used for qRT-PCR analysis are listed in Appendix A. The GenStar kits (2 × RealStar Green Fast Mixture Cat#A301-10) were used to perform fluorescence quantification. The reference gene β-actin was used to normalize the expression values. Each experimental group was performed in triplicate. qRT-PCR data were calculated using the 2^−ΔΔCt^ relative quantification method.

### 2.9. Calculations and Statistical Analysis

The SPSS 22.0 software was used to analyze the data. At the *p* < 0.05 level, the main effect was tested by means of one-way ANOVA. The Duncan *t*-test was used for pairwise comparisons among groups. Results are presented as mean ± standard deviation, and the results are presented in GraphPad Prism 6.0.

## 3. Results

### 3.1. Transcriptomic and 16S rRNA Analysis in Feeding Experiments

Differentially expressed gene (DEG) analysis revealed 362 DEGs upregulated and 513 DEGs downregulated in the TDTGP-4 group (Figure 1A) compared to the blank control group. GO enrichment analysis revealed 43 DEGs upregulated and 50 DEGs downregulated in the immune system progression term and 3 DEGs upregulated and 4 DEGs downregulated in the antioxidant activity term in the TDTGP-4 group compared to the blank control group (Appendix A). Analysis of the abundance of microbes in the intestine of the TDTGP-4 group and the blank control group showed that the abundance of *Vibrionaceae* in the intestine of the TDTGP-4 group decreased relatively, and the abundance of *Rhodobacteraceae* in the intestine of the TDTGP-4 group increased (Figure 1B).

### 3.2. Ammonia Exposure Test

#### 3.2.1. The Change in Hemolymph Factors in the Ammonia Exposure Test

After 72 h of ammonia stress treatment, the hemolymph of the ammonia stress control group had statistically greater ACP content and PO activity than in those of the blank control group (*p* < 0.05) (Figure 2). The AT2 to AT8 groups had statistically greater ACP content, and PO and i-NOS activities, than those of the ammonia stress control and blank control groups (*p* < 0.05). The AT8 group had statistically greater T-AOC content in the hemolymph than the ammonia stress group (*p* < 0.05). The AT2, AT4, and AT8 groups had statistically lower SOD activity in the hemolymph than the T4 group (*p* < 0.05).

#### 3.2.2. Hepatopancreas RNA-Seq Transcriptome Analysis in the Ammonia Exposure Test

To analyze the changes in differential gene expression (DGE) among the groups, the sequencing results of the blank control group, TDTGP-4 group, ammonia stress group, and TDTGP-4 + ammonia stress group were analyzed jointly. PCA showed that, between the TDTGP-4 + ammonia stress group and TDTGP-4, there was a difference in gene expression (Figure 3A). The volcano plot showed 378 DEGs were upregulated and 441 DEGs were downregulated in the ammonia stress group compared to the blank control group. Compared with the ammonia stress group and TDTGP-4 group, a total of 80 and 226 DEGs were upregulated, and 100 and 207 DEGs were downregulated, in TDTGP-4 + ammonia stress group, respectively (Figure 3B–D). GO enrichment analysis showed that the GO terms of each group were similar (Appendix A). Compared with the blank control group, there are 23 DEGs upregulated and 38 DEGs downregulated in the immune system progression term of the ammonia stress group. In terms of antioxidant activity, two DEGs were found to be upregulated and six DEGs were found to be downregulated.

A total of 12 DEGs related to immunity and antioxidation were screened, mainly including phenoloxidase-activating factor 2-like (PPAF2), heme oxygenase (HO), heat shock protein 90b1 (HSP90b1), GTP cyclohydrolase 1-like (cat-4), and putative thioredoxin-like protein 1 (TXNL1) (Figure 4).

Heat map analysis of the selected DEGs showed that the ammonia-stressed control group had statistically lower mRNA expression of PPAF2, Pu, and cat-4, and higher mRNA expression of copper transport protein ATOX1-like (ATOX1), NADPH cytochrome P450 reductase-like (Cpr), TXNL1, HO, and aspartate aminotransferase cytoplasmic-like (GOT1), than the blank control. The AT4 group had statistically greater mRNA expression of PPAF2, Pu, cat-4, Hsp90b1, Chia, and CHIT1 and lower mRNA expression of GOT1 than the ammonia-stressed control group. The mRNA expression of ATOX1, Cpr, TXNL1, and HO of the AT4 group was not different than that of the ammonia-stressed control group. The AT4 group had statistically lower mRNA expression of PPAF2 than the T4 group.

#### 3.2.3. Gut Microbial Composition in the Ammonia Exposure Test

There was a difference between the ammonia stress group and the other three groups, according to principal coordinate analysis (PCoA) (Figure 5A). The HSD analysis and species distribution river map showed that the TDTGP-4 group had statistically significantly lower abundance of *Vibrionaceae* in the gut than the other groups at the family level (*p* < 0.05), and the ammonia stress group had statistically significantly lower abundance of *Flavobacteriaceae* than the other groups (*p* < 0.05). The ammonia stress group and TDTGP-4 + ammonia stress group had significantly lower abundance of *Rhodobacteraceae* than the blank control group (*p* < 0.05) (Figure 5B,C). There was a negative relationship among *Rhodobacteraceae*, as well as *Flavobacteriaceae* and *Vibrionaceae*, in the species correlation network diagram at the family level (Figure 5D).

Alpha diversity analysis showed that there was no statistical difference in Richness, Chao, and Shannon indices between the groups. The TDTGP-4 group had statistically greater Shannon and Simpson indices than the other groups (*p* < 0.05). The TDTGP-4 + ammonia stress group had statistically lower Shannon and Simpson indices than the TDTGP-4 group (*p* < 0.05) (Table 2).

### 3.3. Nitrite Exposure Test

#### 3.3.1. Changes in the Hemolymph Factors in Nitrite Exposure Test

After 72 h of nitrite stress treatment, the nitrite-stressed control had statistically greater ACP levels, and PO and SOD activities, than the blank control group (*p* < 0.05) (Figure 6). The NT2 to NT8 groups had statistically greater ACP levels and lower PO activity than the nitrite-stressed control group (*p* < 0.05). The NT2 and NT4 groups had statistically greater activity of i-NOS than the nitrite-stressed control group (*p* < 0.05). The NT4 and NT8 groups had statistically higher activity of SOD than the nitrite stress control group (*p* < 0.05).

#### 3.3.2. RNA-Seq Transcriptome Analysis of the Hepatopancreas in a Nitrite Exposure Test

PCA showed that there were differences in gene expression composition between the nitrite stress group, TDTGP-4 group, and blank control group, but the DGE composition of the TDTGP-4 + nitrite stress group was similar to that of the TDTGP-4 group (Figure 7A). The volcano plot showed that 136 DEGs were upregulated and 329 DEGs were downregulated in the nitrite stress group compared with the blank control group. Compared with the nitrite stress group, there were 27 DEGs upregulated and 54 DEGs downregulated in the TDTGP-4 + nitrite stress group. Furthermore, 75 DEGs were upregulated and 71 DEGs were downregulated in the TDTGP-4 + nitrite stress group compared to the TDTGP-4 group (Figure 7B–D).

GO Enrichment analysis revealed more DEGs of the immune system processes term and the antioxidant activity term in the nitrite stress group than in the blank control group (upregulated by 14 and 0 and downregulated by 26 and 2, respectively) (Appendix A). The DEGs related to antioxidation and immunity were analyzed by a heatmap (Figure 8). The nitrite stress and NT4 groups had statistically lower mRNA expression of peritrophin-1-like protein (PT-1), PPAF1, possible chitinase 10 (PCHIT10), cat-4, and Pu than the blank control group (Figure 8). The NT4 group had statistically lower mRNA expression of vascular endothelial growth factor D-like (VEGF), α-2 macroglobulin (A2ML1), glutathione peroxidase (GPx), and nicotinamide adenine dinucleotide-dependent enzyme (Sirt1), and higher mRNA expression of Pu, cat-4, HO, and Hsp90b1, than the nitrite stress group. The NT4 group had statistically lower mRNA expression of TXNL1 than the TDTGP-4 group.

#### 3.3.3. Intestinal Microbial Composition in Nitrite Exposure Test

PCoA showed that between the nitrite stress group and the blank control group there was a difference in gut microbial composition. The TDTGP-4 group was statistically different compared to the TDTGP-4 + nitrite-stressed group (Figure 9A). The species distribution river map and the result of HSD analysis showed that the TDTGP-4 group and nitrite stress group had significantly lower abundance of *Vibrionaceae* than the blank control group (*p* < 0.05). The relative abundance of *Rhodobacteraceae* and *Flavobacteriaceae* was not significantly different from that of the other groups (*p* > 0.05) (Figure 9B,C). As shown in the species correlation network diagram, there was a negative correlation between *Rhodobacteraceae*, *Flavobacteriaceae*, and *Vibrionaceae* at the family level (Figure 9D).

Alpha diversity analysis showed that the TDTGP-4 group had statistically higher Richness and Chao indices than the nitrite stress group (*p* < 0.05). The TDTGP-4 group had a significantly higher Shannon index than the blank control group (*p* < 0.05). The nitrite stress group had significantly lower Richness than the blank control group and the TDTGP-4 group (*p* < 0.05) (Table 3).

### 3.4. Quantitative Real-Time PCR Validation

The DEGs of Sirt1, fdxr, HO, Hsp90b1, Pu, GOT1, and VEGF were selected for qPCR verification (Figure 10). The results showed that the qPCR results were generally consistent with the trends of the sequencing results, with differences in individual genes in only a few groups (VEGF in the NT4 group and Pu in the T4 group), probably due to large differences in gene expression between individual *L. vannamei*, but the consistency of the trends in the other data was sufficient to prove that the sequencing results were reliable.

## 4. Discussion

### 4.1. Modulation of Immune and Antioxidant Indicators by the TDTGP in the L. vannamei

Ammonia and nitrite are important metabolites in the culture of *L. vannamei*. They can reach a maximum of 46 mg/L and 20 mg/L, respectively, in the culture water [2,3]. Moderate levels of ammonia and nitrite stress will cause an increase in the immune and antioxidant capacity of the shrimp, but prolonged or high levels of ammonia and nitrite stress will cause an increase and then a decrease in the immune and antioxidant function of the shrimp [15]. In this experiment, changes in hemolymph PO, i-NOS, ACP, SOD, and T-AOC indicators were measured to assess changes in immune and antioxidant function of *L. vannamei*. Among these indices, the SOD and T-AOC in hemolymph were closely related to the antioxidant capacity of *L. vannamei*. SOD has an antioxidant effect and can scavenge superoxide anion radicals (O^2−^) that are harmful to the organism [19]. T-AOC is the total antioxidant content of different antioxidant substances and antioxidant enzymes, such as antioxidant enzymes and vitamin C. This is used to evaluate the antioxidant capacity of bioactive substances [20]. Inducible nitric oxide synthase (i-NOS) induces the conversion of l-arginine to nitric oxide (NO). NO is essential for the inflammatory response and the innate immune system, helping to fight off invading pathogens [21]. However, high levels of NO due to overexpression or dysregulation of i-NOS can lead to toxic effects, including infectious shock, pain, and cancer [22]. Activated PO catalyzes the progressive oxidation of phenols to pathogen-inactivating quinones [1]. During phagocytosis of hemolymph, phagocytic lysosomes act as bactericides by releasing ACP [23]. Therefore, PO, i-NOS, ACP, SOD, and T-AOC can be used as indicators of the immunity and antioxidant capacity of *L. vannamei*.

Plant extracts can modulate the immune and antioxidant function of *L. vannamei* [24]. In this experiment, immune and antioxidant indices such as PO, ACP, i-NOS, SOD, and T-AOC in the hemolymph of the ammonia and nitrite stress groups showed different degrees of increase after ammonia and nitrite stress, indicating that moderate environmental stress can improve the immune function of *L. vannamei*. Furthermore, we observed that the TDTGP group had higher immune parameters such as PO, ACP, i-NOS, SOD, and T-AOC than the ammonia and nitrite stress groups after ammonia stress and nitrite stress, which was similar to previous studies and again confirmed that TDTGP can improve the immunity of *L. vannamei* to some extent [15]. This may be related to the composition of TDTGP. Some studies have reported that aqueous root extract of *Asparagus cochinchinensis* (Lour.) Merr. can increase SOD, CAT, and i-NOS activities in the blood of mice, thus improving their antioxidant capacity [25]. *Panax notoginseng* extract (PNE) [26] can increase SOD activity and T-AOC content in the liver of hybrid grouper. These studies have shown that both aqueous root extract of *Asparagus cochinchinensis* (Lour.) Merr. and *Panax notoginseng* extract have better immune-enhancing and antioxidant effects, which is consistent with our experimental results.

### 4.2. Modulatory Effects of TDTGP on Immune and Antioxidant-Related Genes Expression

The genome has shown significant differential expression in tissues such as the hepatopancreas and intestine of shrimps after Cu, ammonia, or heat stress treatment with environmental factors. For example, Duan et al. [4] found that the decreased expression of immune-related genes and the disturbance of intestinal metabolism in the intestine of *L. vannamei* were induced by ammonia and heat stress. Guo et al. [27] confirmed that the ROS levels in the hemolymph of *L. vannamei* treated with Cu stress were increased, while Cu–Zn SOD and CAT played an important role in protecting against Cu stress. Furthermore, in this study, the differential expression of PPAF2, HO, TXNL1, GCH1L, cat-4, and other genes was observed after TDTGP, ammonia stress, and nitrite stress treatments. Among these genes, prophenoloxidase activating factors (PPAFs) can be translated into a group of clip domain serine proteinases. These proteinases can convert pro-phenoloxidase (pro–PO) to the active form phenol oxidase (PO), which induces phenol oxidase cascade reactions to produce o-diquinone and melanin, and the o-diquinone and melanin can inhibit or kill the pathogens [28]. Heme oxygenase–1 (HO–1) degrades heme (a potent oxidant) to carbon monoxide, bilirubin (an antioxidant derived from biliverdin), and iron. HO–1 is upregulated during oxidative stress and can help protect cells and tissue from oxidative stress [29]. TXNL1 may be one of the thiooxenites (TRX) in *L. vannamei*. TRXs are known to promote hypoxia inducible factor-1alpha (HIF-1α) expression and activity, and epidermal growth factor expression [30,31].

In the present experiment, ammonia stress and nitrite stress could, to some extent, induce a decrease in the expression of some antioxidant– and immune–related genes after ammonia and nitrite stress. Compared with the blank control group, the mRNA expression of Pu, cat-4, and PPAF2 in the hepatopancreas was decreased in the ammonia and nitrite stress groups. Of course, there were some antioxidant- and immune-related genes with increased expression, such as TXNL1 and HO. This finding is similar to that of Duan [4], who found that the expression of immune genes was reduced by heat and ammonia stress. After TDTGP treatment, the gene expressions of PPAF2, Pu, cat-4, Hsp90b1, and Chia were higher in the AT4 group than in the ammonia-stressed group, and the gene expressions of Pu, cat-4, Hsp90b1, and HO were higher in the NT4 group than in the nitrite-stressed control group. It was shown that TDTGP treatment can reduce the effects of ammonia stress and nitrite stress on the expression of genes such as PPAF2, Pu, cat-4, Hsp90b1, and Chia in hepatopancreas and maintain their stability.

### 4.3. Effect of TDTGP on Gut Microbiome

The gut microbiome of marine creatures is directly linked to their environment. *Flavobacteriaceae*, *Vibrio*, and *Rhodobacteraceae* are widely found in the ocean and in the guts of marine creatures such as sea cucumbers and *L. vannamei* [32]. Among these, the various organic compounds in seawater were directly or indirectly mineralized, the production of carbohydrates was increased, and the rate of intestinal regeneration of sea cucumbers was promoted, which may be induced by the enzymatic ability of *Flavobacteriaceae* [33]. *Rhodotoraceae* are aquatic photosynthetic bacteria and can be used as probiotics. The high abundance of *Rhodotoraceae* (including some *Ruegeria taxa*) may play an active role in the promotion of digestion, the provision of nutrients, and the inhibition of pathogens [32]. These studies have shown that Rhodobacteraceae and *Flavobacteriaceae* may play a key role in maintaining the stability of the community structure during the regeneration of the gut [34]. In addition, one of the major causes of shrimp mortality is *Vibrio* spp. infection, including V. *parahaemolyticus*, V. *anguillarum*, and V. *splendidus*. Among them, V. *parahaemolyticus* causes acute hepatopancreatic necrosis disease (AHPND) [35,36]. Several medicinal plants appeared capable of inhibiting the growth of *Vibrio* pathogens in vivo or in vitro, such as *Syzygium cumini* (*Myrtaceae* family), *Rhodomyrtus tomentosa*, and *Psidium guajava*, with significant activity against *Vibrio*.

The gut microbiome is moderated by plant extracts. For example, treatment with *Panax notoginseng* saponins (PNSs) shaped the murine gut microbiome by increasing the abundances of *Akkermansia muciniphila* and *Parabacteroides distasonis* [37]. This may be due to the fact that the *Panax notoginseng* saponins have low drug permeability, resulting in poor intestinal absorption into the body, and are therefore able to interact with the intestinal microbiota for a longer period of time, thereby influencing the gut microbial ecosystem [37,38]. A neutral polysaccharide of *Asparagus cochinchinensis* (Lour.) Merr. (ACNP) with an apparent molecular weight of 2460 Da was purified from asparagus by Sun et al. It was found that the ACNP could be digested by intestinal microbiota. Subsequently, the pH was significantly decreased and the levels of total short-chain fatty acids, acetic acid, propionic acid, valeric acid, and valeric acid were significantly increased in fecal culture. The composition of the gut microbiota and the consumption of *Haemophilus* were regulated by ACNP by stimulating the growth of *Prevoteella*, *Megalomonas*, and *Bifidobacterium* [39].

In the present study, there was an increase in the abundance of *Rhodobacteraceae* and a decrease in the abundance of *Vibrionaceae* after TDTGP treatment in the intestine of *L. vannamei*. Furthermore, in vitro TDTGP inhibition tests were performed and showed no direct inhibition of *Vibrio* by TDTGP (Appendix A), and network graph analysis revealed that *Rhodobacteraceae* and *Flavobacteriaceae* showed antagonistic effects with *Vibrionaceae* abundance (Figure 5D and Figure 9D). Therefore, the reason may be that the polysaccharides of *Asparagus cochinchinensis* (Lour.) Merr. and the total saponins of *Panax notoginseng* (Burkill) F.H. Chen ex C.H. in TDTGP were digested and degraded by the intestinal microbiota, which promoted the proliferation of *Rhodobacteraceae* and *Flavobacteriaceae* in the intestine. The large accumulation of *Rhodotoraceae* and *Flavobacteriaceae* may promote regeneration and digestion in the intestine of *L. vannamei* and play an active role in providing nutrients and inhibiting pathogens. These results are similar to those of Yanbing Qiao [40] and others who showed that feeding different doses of β-glucan for 35 days significantly reduced the relative abundances of *Vibrio*, *Rheinheimera*, and *Demequina*, and the relative abundance of *Lacrobacillus* was significantly higher. Nevertheless, the abundance of *Rhodobacteraceae* decreased and that of *Vibrionaceae* increased in the AT4 group. These results are similar to those of Duan [4], who found that the abundance of *Formosa*, *Kriegella*, *Ruegeria*, *Rhodopirellula*, and *Lutimonas* decreased and the abundance of pathogenic bacteria such as *Vibrio* increased under heat and ammonia stress.

Several studies have shown that gut microbiota are closely related to host immune function and antioxidant capacity; for example, gut microbiota enhance host antioxidant capacity through the generation of reactive sulfur species [32]. This suggests that the gut microbiome and their metabolites may contribute to host antioxidant capacity and immune function. It has been shown that pumpkin juice fermented by *Rhodobacter sphaeroides* can improve the antioxidant capacity of pumpkin juice in vitro and increase the stability of the gut microbiome in mice [41]. Carotenoids with antioxidant activity can be produced by *Flavobacteriaceae* [42], suggesting that the by-products of both bacteria may secrete substances with antioxidant activity that may modulate host immune functions to some extent. In this study, there was an increase in the abundance of *Rhodotoraceae* and *Flavobacteriaceae* in the gut and some increase in immune function after TDTGP treatment, but the direct relationship between the two needs to be further investigated and established.

## 5. Conclusions

In conclusion, TDTGP treatment increased PO, SOD, and i-NOS activity, and ACP and T-AOC levels, in the hemolymph of *L. vannamei* and improved the shrimp’s resistance to ammonia and nitrite stress. Transcriptome sequencing revealed that TDTGP treatment reduced the effect of ammonia stress and nitrite stress on the expression of Pu, cat-4, PPAF2, HO, and Hsp90b1, which may be the basis for TDTGP to increase resistance to ammonia and nitrite stress in *L. vannamei*. Gut microbiome sequencing revealed that environmental stress induced gut microbiome dysbiosis and increased the abundance of harmful bacteria such as *Vibrionaceae*. TDTGP treatment reduced the abundance of *Vibrionaceae* and other bacteria and increased the stability of the gut microbiome, and *Rhodobacteraceae* and *Flavobacteriaceae* may be the marker microbiomes after TDTGP.

## Figures and Tables

**Figure 1 antioxidants-12-01262-f001:**
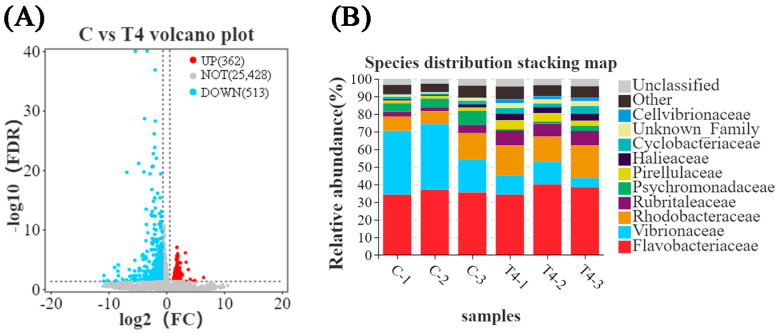
The results of transcriptome and 16s RNA in feeding experiments. Notes: C: blank control group; T4: TDTGP-4 group. (**A**) Volcano plot of DEGs. Red dots indicate significantly elevated expression between the two samples, green dots indicate significantly decreased expression between the two samples, and gray dots indicate no significant differential expression. (**B**) Species distribution stacking map in the blank control group and TDTGP-4 group.

**Figure 2 antioxidants-12-01262-f002:**
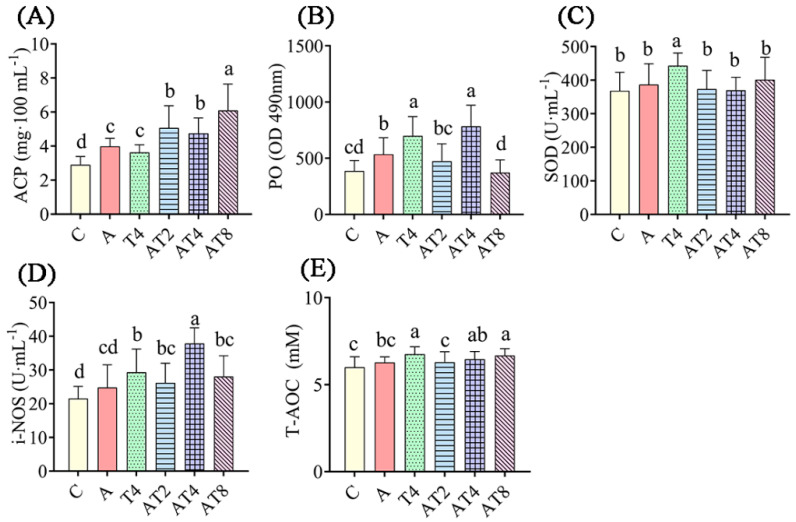
The changes in enzyme activity in the ammonia exposure test. Notes: C: blank control group; A: ammonia stress group; AT2 to AT8: TDTGP-2 + ammonia stress group (AT2) to TDTGP-8 + ammonia stress group (AT8), the same below. (**A**–**E**): The contents of T-AOC, ACP, and activities of PO, SOD, i-NOS, and in hemolymph, separately. a–d Letters with different superscripts are statistically different (*p* < 0.05).

**Figure 3 antioxidants-12-01262-f003:**
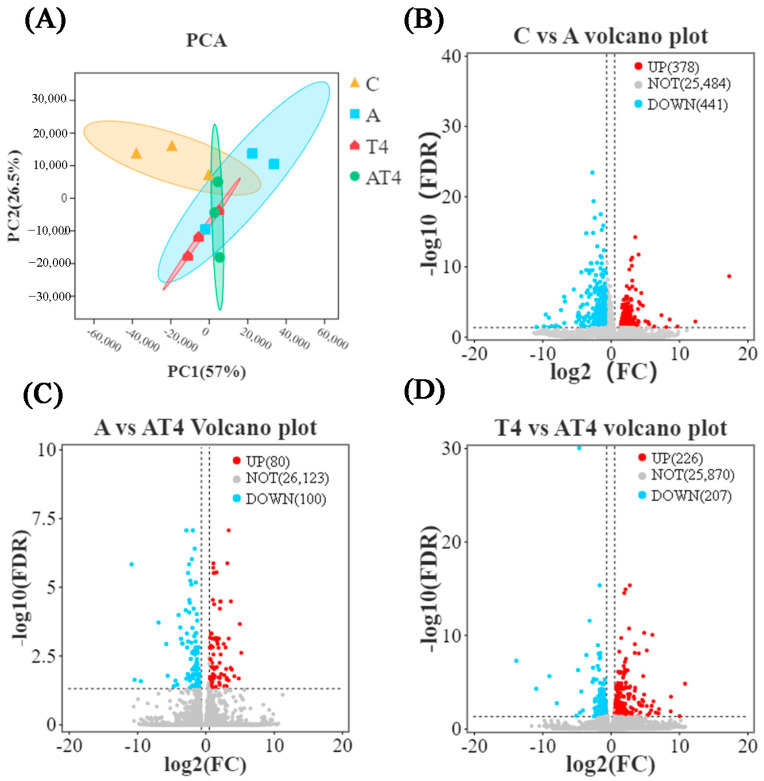
The results of RNA-Seq transcriptome in the ammonia exposure test. Notes: (**A**): The principal component analysis (PCA) results with each group in the hepatopancreas RNA-Seq transcriptome. (**B**–**D**): Volcano plot highlighting significantly expressed genes.

**Figure 4 antioxidants-12-01262-f004:**
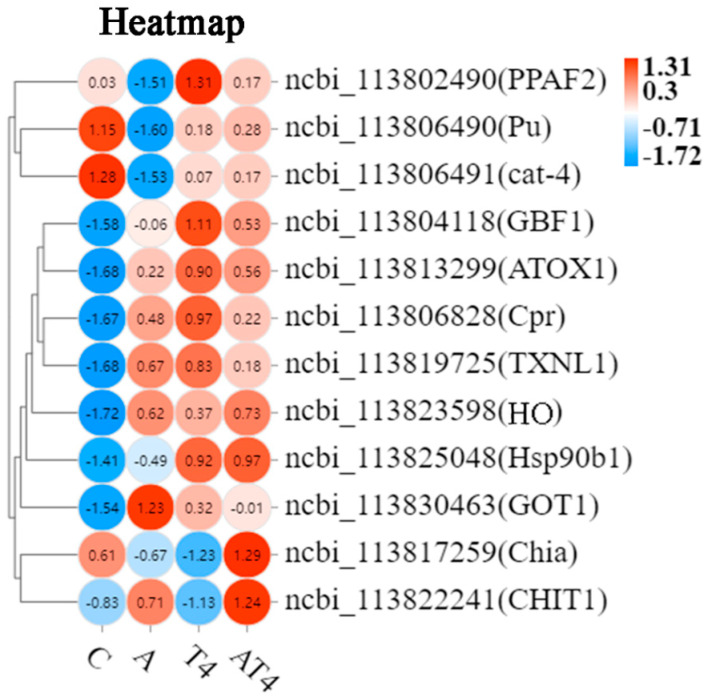
The heatmap of the differentially expressed genes related to antioxidation and immunity in each group. Notes: White color indicates no relative expression, red color indicates elevated relative expression, and blue color indicates decreased differential expression.

**Figure 5 antioxidants-12-01262-f005:**
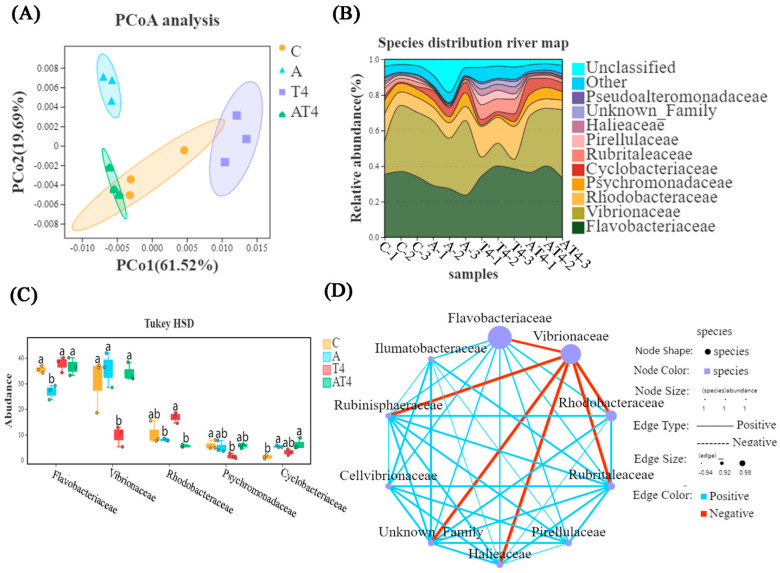
Analysis of changes in intestinal microorganisms. Notes: (**A**) The principal coordinate analysis (PCoA) was carried out with the intestinal microbiota of the C, A, T4, and AT4 groups. (**B**) The species distribution river map of the abundance of the gut microbiota. (**C**) The HSD analysis (*p* < 0.05) was carried out on the top five most abundant gut microbiotas at the family level. (**D**) The species correlation network plot at the family level in the ammonia stress test.

**Figure 6 antioxidants-12-01262-f006:**
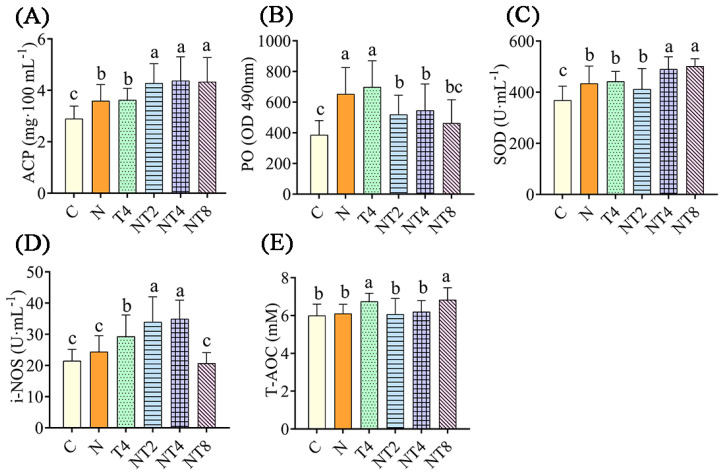
The changes in enzyme activity in hemolymph in the nitrite exposure test. Notes: C: blank control group; N: nitrite stress group; NT2 to NT8: TDTGP-2 + nitrite stress group (NT2) to TDTGP-8 + nitrite stress group (NT8), the same below. (**A**–**E**): The activities of SOD, PO, and i-NOS, and contents of ACP, T-AOC in hemolymph, separately. a, b, c: Letters with different superscripts are statistically different (*p* < 0.05).

**Figure 7 antioxidants-12-01262-f007:**
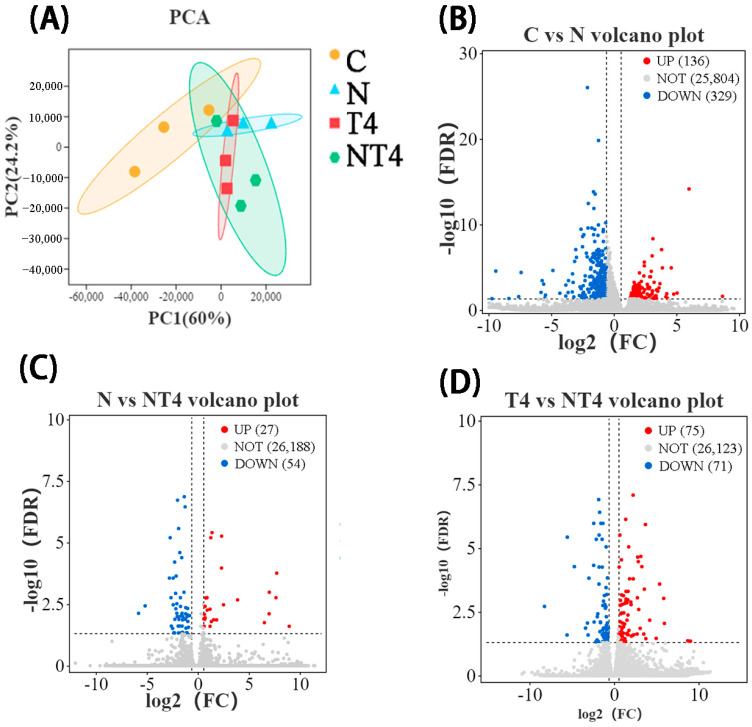
The results of RNA-Seq transcriptome in the nitrite exposure test. Notes: (**A**): The principal component analysis (PCA) results with each group in the hepatopancreas RNA-Seq transcriptome. (**B**–**D**): Volcano plot highlighting significantly expressed genes.

**Figure 8 antioxidants-12-01262-f008:**
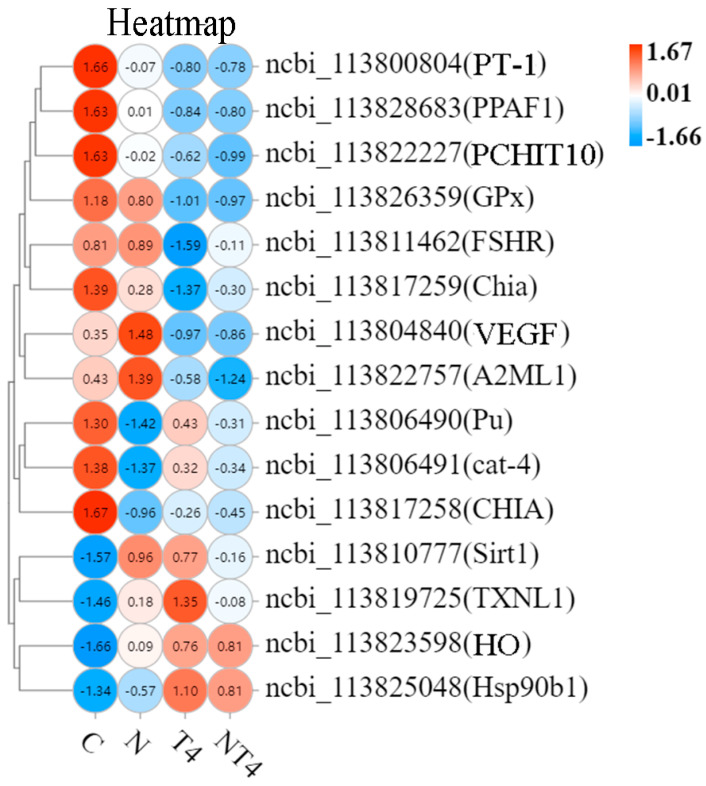
The heatmap of the differentially expressed genes related to antioxidation and immunity in each group in the nitrite exposure test. Notes: Red color indicates elevated relative expression, white color indicates no relative expression, and blue color indicates decreased differential expression.

**Figure 9 antioxidants-12-01262-f009:**
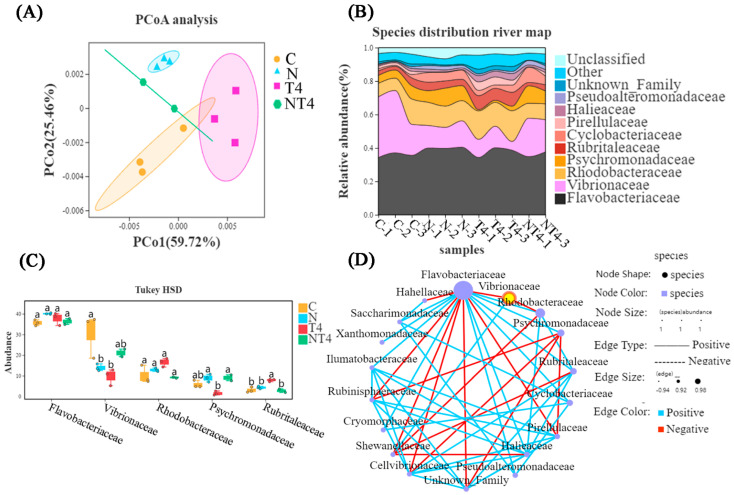
Analysis of the changes in intestinal microorganisms in nitrite exposure test. Notes: (**A**) The principal coordinate analysis (PCoA) was carried out with the intestinal microbiota of the C, N, T4, and NT4 groups. (**B**) The species distribution river map of the abundance of the gut microbiota. (**C**) The HSD analysis (*p* < 0.05) was carried out on the top five most abundant gut microbiotas at the family level. (**D**) The species correlation network plot at the family level in the nitrite stress test. a, b: Letters with different superscripts are statistically different (*p* < 0.05).

**Figure 10 antioxidants-12-01262-f010:**
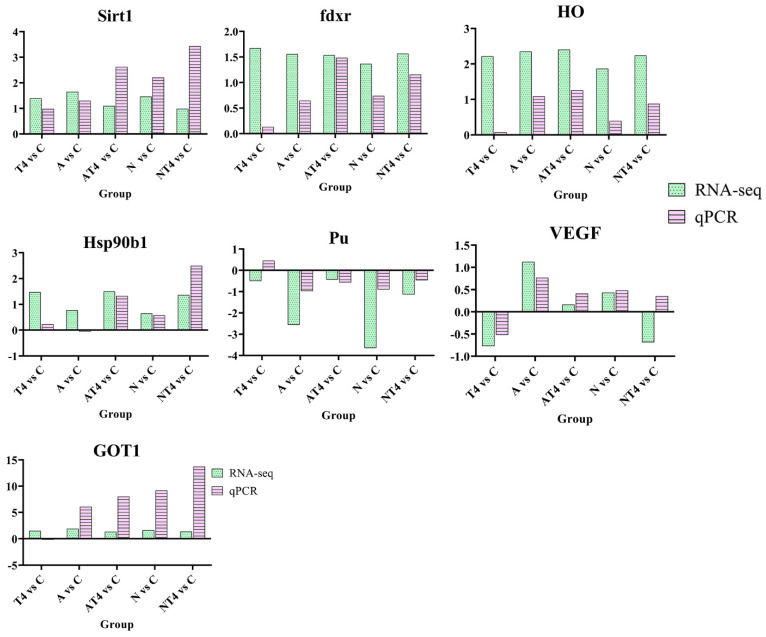
Validation of RNA-seq data by qRT-PCR. Notes: C: blank control group; A: ammonia stress group; N: nitrite stress group; T4: TDTGP-4 group; NT4: TDTGP-4 + nitrite stress group; AT4: TDTGP-4 + ammonia stress group. Green bar: RNA-seq; purple bar: qRT-PCR. The expression levels of all genes are normalized to the corresponding level of beta-actin in the qRT-PCR. The data from RNA-seq represent the log2 fold change in the expression in the different groups in comparison to the blank control group.

**Table 1 antioxidants-12-01262-t001:** Formula (%) and proximate composition (%) of the trial diets.

	Groups			
	Blank Control Group and Stress Control Group	TDTGP-2 Group	TDTGP-4 Group	TDTGP-8 Group
Ingredients				
Fish meal	20.00	20.00	20.00	20.00
Wheat gluten	30.00	30.00	30.00	30.00
Wheat meal	20.00	20.00	20.00	20.00
Cellulose	18.00	17.80	17.60	17.20
Fish oil	2.50	2.50	2.50	2.50
Soybean oil	2.50	2.50	2.50	2.50
Soybean phospholipids	2.00	2.00	2.00	2.00
Gelatin	2.00	2.00	2.00	2.00
Choline chloride	1.00	1.00	1.00	1.00
Vitamin mix	1.00	1.00	1.00	1.00
Mineral mix	1.00	1.00	1.00	1.00
TDTGP		0.20	0.40	0.80
Proximate composition (as fed)				
Crude protein	43.34	43.06	42.75	42.01
Crude fat	7.31	7.26	7.12	7.04
Crude ash	12.91	12.98	13.05	13.08

Fish meal: 69.52% crude protein, 8.02% crude fat; wheat glutens: 78.59% crude protein; 0.121% crude fat; wheat meal: 16.15% crude protein, 1.07% crude fat; fish oil: 99.00% crude fat; soybean oil: 95.00% crude fat; soybean phospholipids: 7.00% crude protein, 40.00% crude fat; of dry matter. Vitamin mixture (%): riboflavin, 3.362; thiamine, 1.868%; vitamin K3, 0.747%; inositol, 59.773%; pyridoxine hydrochloride, 1.494%; vitamin B12, 0.007%; calcium pantothenate, 4.483%; biotin,0.097%; vitamin A, 2.391%; vitamin D, 0.374%; nicotinic acid, 14.943%; folic acid, 1.494%; vitamin E, 8.966%. Mineral mixture (%): KI, 0.023%; NaF, 0.057%; Fe_2_(SO_4_)_3_, 2.287%; ZnSO_4_, 1.572%; CoCl_2_·6H_2_O, 1.429%; CuSO_4_·5H_2_O, 0.286%; MgSO_4_, 5.718%; NaCl, 2.859%; Ca(H_2_PO_4_)_2_, 85.768%.

**Table 2 antioxidants-12-01262-t002:** Alpha diversity of *L. vannamei* gut microbiota after ammonia stress.

Group	Richness	Chao	Shannon	Simpson	Chao (%)
Blank control group	329.67 ± 55.97	353.93 ± 57.03	4.55 ± 0.49 ^b^	0.90 ± 0.04 ^b^	99.96 ± 0.01
Ammonia stress group	305.00 ± 82.15	349.18 ± 82.52	4.73 ± 0.14 ^b^	0.92 ± 0.01 ^ab^	99.96 ± 0.01
TDTGP-4 group	354.33 ± 7.51	379.18 ± 8.60	5.42 ± 0.13 ^a^	0.95 ± 0.01 ^a^	99.96 ± 0.001
TDTGP-4 + ammonia stress group	285.00 ± 39.89	317.27 ± 46.60	4.30 ± 0.03 ^b^	0.89 ± 0.001 ^b^	99.97 ± 0.02

^a, b^ Letters with different superscripts are statistically different (*p* < 0.05). Data are expressed as mean ± SD. Same table below.

**Table 3 antioxidants-12-01262-t003:** Alpha diversity analysis of intestinal microbial in *L. vannamei* in the nitrite stress test.

Group	Richness	Chao	Shannon	Simpson	Coverage (%)
Blank control group	312.00 ± 39.89 ^ab^	326.12 ± 39.34 ^ab^	4.53 ± 0.49 ^b^	0.90 ± 0.04	99.97 ± 0.01 ^a^
Nitrite stress group	245.00 ± 6.56 ^c^	271.44 ± 4.97 ^b^	4.91 ± 0.08 ^ab^	0.94 ± 0.01	99.96 ± 0.01 ^ab^
TDTGP-4 group	354.67 ± 10.02 ^a^	376.85 ± 13.49 ^a^	5.44 ± 0.14 ^a^	0.95 ± 0.01	99.96 ± 0.01 ^ab^
TDTGP-4 + nitrite stress group	289.50 ± 53.03 ^bc^	320.27 ± 46.64 ^ab^	4.92 ± 0.37 ^a^	0.94 ± 0.01	99.96 ± 0.001 ^b^

^a, b, c^: Letters with different superscripts are statistically different (*p* < 0.05).

## Data Availability

Some or all data, models, or code generated or used in the study, are available from the corresponding author by request. The data are not publicly available due to privacy.

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
