# Peer review of "RNA-Seq and 16S rRNA Reveals That Tian–Dong–Tang–Gan Powder Alleviates Environmental Stress-Induced Decline in Immune and Antioxidant Function and Gut Microbiota Dysbiosis in Litopenaeus vannami"

_antioxidants, 2023, doi:10.3390/antiox12061262_

Round 1

Reviewer 1 Report

Comments and Suggestions for Authors

: In this study, Tian-Dong-Tang-Gan Powder (TDTGP) was added to the feed of Pacific white shrimp, and its effect on ammonia and nitrite stress and intestinal microbial changes were studied. Except for the part of the intestinal microbiome, it is considered that the two studies already written by the authors have already demonstrated the effect of TDTGP on the immunity and stress of shrimp (Xie et al., 2021; Xie et al., 2022). Therefore, it is considered that the difference between this study and the previous studies is the change in intestinal microflora. The fact that TDTGP can enhance the growth of Rhodobacteriaceae and Flavobacteriaceae in the gut microbiome and reduce only the Vibrio colony requires further clarification and a nutritional composition table for TDTGP. It is a very interesting result, so I think it will be a good paper if an in-depth consideration is added to this part rather than a short guess.

I send a message of support for the efforts of the authors.

Xie, XiaoDong, et al. "Effects of medical herbs in TianDongTangGan powder on nonspecific immune responses and resistance to acute ammonia stress in Litopenaeus vannamei." Aquaculture Research 52.7 (2021): 3360-3370.

Xie, Xiao-Dong, et al. "Effect of medical herbs in Tian-Dong-Tang-Gan powder on the oxidative stress induced by ammonia and nitrite in Litopenaeus vannamei." Aquaculture 548 (2022): 737584.

Discussion

Page 16, Line 414 - 454

: It is a really interesting result that the distribution of gut microbes can be selectively changed by adding TDTGP. In particular, it is surprising that Vibrio, known as a pathogenic microorganism, is reduced, and that only two other large intestinal microbial communities, Flavobacteriaceae and Rhodobacteriaceaea, enhance proliferation. The author wrote "The reason may be that the polysaccharides of Asparagus cochinchinensis (Lour.) Merr. and the total saponins of Panax notoginseng (Burkill) F. H. Chen ex C. H. in TDTGP were digested and degraded by the intestinal microbiota, which promoted the proliferation of Rhodobacteriaceae. and Flavobacteriaceae in the intestine.", but I think it is difficult to understand with this alone. I read the author's previous papers, but I could not find a clear nutritional composition table for TDTGP, so the nutrition for this intestinal microbial community It is considered necessary for a study on the change in health, and a study on this part and the addition of a nutritional ingredient table are necessary.

Author Response

Comments and Suggestions for Authors

In this study, Tian-Dong-Tang-Gan Powder (TDTGP) was added to the feed of Pacific white shrimp, and its effect on ammonia and nitrite stress and intestinal microbial changes were studied. Except for the part of the intestinal microbiome, it is considered that the two studies already written by the authors have already demonstrated the effect of TDTGP on the immunity and stress of shrimp (Xie et al., 2021; Xie et al., 2022). Therefore, it is considered that the difference between this study and the previous studies is the change in intestinal microflora. The fact that TDTGP can enhance the growth of Rhodobacteriaceae and Flavobacteriaceae in the gut microbiome and reduce only the Vibrio colony requires further clarification and a nutritional composition table for TDTGP. It is a very interesting result, so I think it will be a good paper if an in-depth consideration is added to this part rather than a short guess.

I send a message of support for the efforts of the authors.

  Xie, Xiao‐Dong, et al. "Effects of medical herbs in Tian‐Dong‐Tang‐Gan powder on non‐specific immune responses and resistance to acute ammonia stress in Litopenaeus vannamei." Aquaculture Research 52.7 (2021): 3360-3370.

Xie, Xiao-Dong, et al. "Effect of medical herbs in Tian-Dong-Tang-Gan powder on the oxidative stress induced by ammonia and nitrite in Litopenaeus vannamei." Aquaculture 548 (2022): 737584.

Question : Discussion  Page 16, Lines 414 - 454

It is a really interesting result that the distribution of gut microbes can be selectively changed by adding TDTGP. In particular, it is surprising that Vibrio, known as a pathogenic microorganism, is reduced, and that only two other large intestinal microbial communities, Flavobacteriaceae and Rhodobacteriaceaea, enhance proliferation. The author wrote "The reason may be that the polysaccharides of Asparagus cochinchinensis (Lour.) Merr. and the total saponins of Panax notoginseng (Burkill) F. H. Chen ex C. H. in TDTGP were digested and degraded by the intestinal microbiota, which promoted the proliferation of Rhodobacteriaceae. and Flavobacteriaceae in the intestine.", but I think it is difficult to understand with this alone. I read the author's previous papers, but I could not find a clear nutritional composition table for TDTGP, so the nutrition for this intestinal microbial community It is considered necessary for a study on the change in health, and a study on this part and the addition of a nutritional ingredient table are necessary.

Answer: The nutritional composition of the TDTGP has been supplemented and the results of the intestinal microbiota section have been discussed in more detail in Page 2, L67-78 of the revised manuscript:

TDTGP is composed of polysaccharides of Asparagus cochinchinensis (Lour.) Merr. (84.08% total polysaccharide purity, 3.1% moisture), total saponins of Panax notoginseng (Burkill) F.H. Chen ex C.H. (mainly composed of 7.20% notoginsenoside R1, 29.39% ginsenoside Rg1, 4.29% ginsenoside Re, 28.38% ginsenoside Rb1, and 6.77% ginsenoside RD for a total of 76.03%) were mixed with silica (Zhengzhou Yuanze Chemical Co., Zhengzhou, China) in a ratio of 5: 1: 2. The polysaccharides of Asparagus cochinchinensis (Lour.) Merr. and the total saponins of Panax notoginseng (Burkill) F. H. Chen ex C. H. were extracted in the same way as in the previous study (16).

Finally, the total polysaccharide content was 56.90%, the total saponin content was 8.93% (0.901% notoginsenoside R1, 3.413% ginsenoside Rg1, 0.481% ginsenoside Re, 3.354% ginsenoside Rb1, and 0.779% ginsenoside RD, respectively), and the content of silica and other substances was 31.17% per 1 g TDTGP.

The changes in Page 16, L437-500 in the revised manuscript::

4.3 Effect of TDTGP on gut microbiome

The gut microbiome of marine creatures is directly linked to their environment. Flavobacteriaceae, Vibrio, and Rhodobacteraceae are widely found in the ocean and in the guts of marine creatures such as sea cucumbers and L. vannamei (30). Among these, the various organic compounds in seawater were directly or indirectly mineralised, the production of carbohydrates was increased, and the rate of intestinal regeneration of sea cucumbers was promoted, which may be induced by the enzymatic ability of Flavobacteriaceae (31). Rhodotoraceae are aquatic photosynthetic bacteria and can be used as probiotics. The high abundance of Rhodotoraceae (including some Ruegeria taxa) may play an active role in the promotion of digestion, the provision of nutrients and the inhibition of pathogens (30). These studies have shown that Rhodobacteraceae and Flavobacteriaceae may play a key role in maintaining the stability of the community structure during the regeneration of the gut (32). In addition, one of the major causes of shrimp mortality is Vibrio spp infection, including V. parahaemolyticus, V. anguillarum, and V. splendidus. Among them, V. parahaemolyticus causes acute hepatopancreatic necrosis disease (AHPND) (33, 34). Several medicinal plants appeared the capable of inhibiting the growth of Vibrio pathogens in vivo or in vitro, such as Syzygium cumini (Myrtaceae family), Rhodomyrtus tomentosa, and Psidium guajava with significant activity against Vibrio.

The gut microbiome is moderated by plant extracts. For example, treatment with Panax notoginseng saponins (PNS) shaped the murine gut microbiome by increasing the abundances of Akkermansia muciniphila and Parabacteroides distasonis (35). This may be due to the fact that the Panax notoginseng saponins have low drug permeability, resulting in poor intestinal absorption into the body, and therefore are able to interact with the intestinal microbiota for a longer period of time, thus to influence the gut microbial ecosystem (35, 36). A neutral polysaccharides of Asparagus cochinchinensis (Lour.) Merr. (ACNP) with an apparent molecular weight of 2460 Da was purified from asparagus by Sun et al. It was found that the ACNP could be digested by intestinal microbiota. Subsequently, the pH was significantly decreased and the levels of total short-chain fatty acids, acetic acid, propionic acid, valeric acid, and valeric acid were significantly increased in faecal culture. The composition of the gut microbiota and the consumption of Haemophilus were regulated by ACNP by stimulating the growth of Prevoteella, Megalomonas, and Bifidobacterium (37).

In the present study, the abundance of Rhodobacteriaceae and Flavobacteriaceae in the intestine of L. vannamei increased, while the abundance of Vibrionaceae decreased after TDTGP treatment. Furthermore, in vitro TDTGP inhibition tests were performed and showed no direct inhibition of Vibrio by TDTGP (Fig. S7); and network graph analysis revealed that Rhodobacteriaceae and Flavobacteriaceae showed antagonistic effects with Vibrionaceae abundance (Fig. 5D and Fig. 9D); therefore, the reason may be that the polysaccharides of Asparagus cochinchinensis (Lour.) Merr. and the total saponins of Panax notoginseng (Burkill) F. H. Chen ex C. H. in TDTGP were digested and degraded by the intestinal microbiota, which promoted the proliferation of Rhodobacteriaceae and Flavobacteriaceae in the intestine. The large accumulation of Rhodotoraceae and Flavobacteriaceae may promote regeneration and digestion in the intestine of L. vannamei and play an active role in providing nutrients and inhibiting pathogens. These results are similar to those of Yanbing Qiao (38) and others who showed that feeding different doses of β-glucan for 35 days significantly reduced the relative abundances of Vibrio, Rheinheimera and Demequina; and the relative abundance of Lacrobacillus was significantly higher. Nevertheless, the abundance of Rhodobacteriaceae decreased and that of Vibrionaceae increased in the AT4 group. These results are similar to those of Duan (4) who found that the abundance of Formosa, Kriegella, Ruegeria, Rhodopirellula, and Lutimonas decreased and the abundance of pathogenic bacteria such as Vibrio increased under heat and ammonia stress.

Several studies have shown that gut microbiota are closely related to host immune function and antioxidant capacity; for example, gut microbiota enhance host antioxidant capacity through the generation of reactive sulfur species (30). This suggests that gut microbiome and their metabolites may contribute to host antioxidant capacity and immune function. It has been shown that pumpkin juice fermented by Rhodobacter sphaeroides can improve the antioxidant capacity of pumpkin juice in vitro and increase the stability of the gut microbiome in mice (39). Carotenoids with antioxidant activity can be produced by Flavobacteriaceae (40), suggesting that the by-products of both bacteria may secrete substances with antioxidant activity that may modulate host immune functions to some extent. In this study, there was an increase in the abundance of Rhodotoraceae and Flavobacteriaceae in the gut and some increase in immune function after TDTGP treatment, but the direct relationship between the two needs to be further investigated and established.

Reviewer 2 Report

1. I suggest that the authors cite some other work recently published in Antioxidants and other journals dealing with natural antimicrobials and their antioxidant role in shrimp.

2. Lane 50. please correct "oral plant extracts", I don't think this is a correct term.

3. section 2.1 in MM needs more detail.

4. Lane 89 - use capital letter for the word stress.

High quality manuscript.

Author Response

Review 2: Comments and Suggestions for Authors

Question 1. I suggest that the authors cite some other work recently published in Antioxidants and other journals dealing with natural antimicrobials and their antioxidant role in shrimp.

Answer: As suggested, research relevant to the content of our study was cited in Page 16,Line 480-488 of the revised manuscript:

These results are similar to those of Yanbing Qiao (38) and others who showed that feeding different doses of β-glucan for 35 days significantly reduced the relative abundances of Vibrio, Rheinheimera and Demequina; and the relative abundance of Lacrobacillus was significantly higher.

Reference:

  1. Qiao Y., Zhou L., Qu Y., Lu K., Han F., Li E. Effects of Different Dietary β-Glucan Levels on Antioxidant Capacity and Immunity, Gut Microbiota and Transcriptome Responses of White Shrimp (Litopenaeus vannamei) under Low Salinity. Antioxidants (Basel). 2022, 11(11). https://doi.org/10.3390/antiox11112282

Question 2. Line 50. please correct "oral plant extracts", I don't think this is a correct term.

Answer: It has been changed to 'Plant extracts' in Page 2, Line 49 of the revised manuscript:

Plant extracts have been shown to regulate host immunity, antioxidant, and gut microbiota composition in several studies.

Question 3. section 2.1 in MM needs more detail.

Answer: More details have been added in section 2.1 in Page 2,L67-78 in the revised manuscript:

TDTGP is composed of polysaccharides of Asparagus cochinchinensis (Lour.) Merr. (84.08% total polysaccharide purity, 3.1% moisture), total saponins of Panax notoginseng (Burkill) F.H. Chen ex C.H. (mainly composed of 7.20% notoginsenoside R1, 29.39% ginsenoside Rg1, 4.29% ginsenoside Re, 28.38% ginsenoside Rb1, and 6.77% ginsenoside RD for a total of 76.03%) were mixed with silica (Zhengzhou Yuanze Chemical Co., Zhengzhou, China) in a ratio of 5: 1: 2. The polysaccharides of Asparagus cochinchinensis (Lour.) Merr. and the total saponins of Panax notoginseng (Burkill) F. H. Chen ex C. H. were extracted in the same way as in the previous study (16).

Finally, the total polysaccharide content was 56.90%, the total saponin content was 8.93% (0.901% notoginsenoside R1, 3.413% ginsenoside Rg1, 0.481% ginsenoside Re, 3.354% ginsenoside Rb1, and 0.779% ginsenoside RD, respectively), and the content of silica and other substances was 31.17% per 1 g TDTGP.

Question 4. Lane 89 - use capital letter for the word stress.

Answer: “2.3. Stress experiment” has been modified in Page 3, L97 of the revised manuscript.

Reviewer 3 Report

The authors evaluated the effects of Tian-Dong-Tang-Gan (TDTGP) on immune and antioxidant function and gut microbiota dysbiosis in Litopenaeus vannami was induced stress by ammonia and nitrite. The results showed that TDTGP could regulate the immunity and antioxidant of L. vannamei by increasing the mRNA expression levels of immunity and antioxidantrelated genes and regulating the abundance of Rhodobacteraceae and Flavobacteriaceae in the gut microbiota. This might be an interesting topic for publication in this journal. I have comments, explained below. I hope that my comments are very useful for the improvement of this research.

Comments

(1)   TDTGP: The components of TDTGP are not described in detail. Please describe in detail the types of constituent polysaccharides and saponins contained in TDTGP.

(2)   Group name: Group names are not unified in the MS. For example, T4 is a mixture of T4 and T-4.

(3)   L53-55: This reference evaluated the effects of green tea polyphenols on mice. I think that studies on crustaceans, including shrimp, should be given as references.

(4)   L69: What is the origin of Silica?

(5)   L89: Capitalize the first letter of “stress experiment”.

(6)   L107: The salinity is shown as 28%, is this correct? It is 10 times denser than normal seawater.

(7)   L113: Did the authors not measure the size of the shrimp after the experiment period? I think this is very important data.

(8)   Section 2.9: It is not indicated which post hoc test was used. Please show post hoc test in this section.

(9)   L173-174: Figure 1B does not contain the Flavobacterium. In the text, is Flavobacterium a mistake for the Flavobacteriaceae.

(10)Figure 2 and 6: Despite the fact that there are not very large differences, there are significant differences among the groups. An explanation of which test method was used is needed (s same as comment 8). Also, is this minute change a physiologically significant change?

(11)Figure 3A: It states that the PCA results showed significant differences between the groups (in L201-203). Is it possible to tell from this figure alone whether there is a significant difference? L278-282 and L312-313 are described in the same way, but without evidence.

(12)L338-339: Is this content true? On the contrary, there are some genes that are fluctuating. More careful explanation is needed.

(13)Discussion: The relationship between gut microbiota and antioxidant capacity and immune function has not been discussed. Please discuss the relationship between gut microbiota and antioxidant capacity and immune function.

(14)Discussion: Please discuss that the relationship between the components of TDTGP and antioxidant capacity and immune function in L. vannami.

(15)L433-435: Ackermania and Arabidopsis thaliana are plants. Please review the text.

Author Response

Review 3:Comments and Suggestions for Authors

The authors evaluated the effects of Tian-Dong-Tang-Gan (TDTGP) on immune and antioxidant function and gut microbiota dysbiosis in Litopenaeus vannami was induced stress by ammonia and nitrite. The results showed that TDTGP could regulate the immunity and antioxidant of L. vannamei by increasing the mRNA expression levels of immunity and antioxidant related genes and regulating the abundance of Rhodobacteraceae and Flavobacteriaceae in the gut microbiota. This might be an interesting topic for publication in this journal. I have comments, explained below. I hope that my comments are very useful for the improvement of this research.

Comments

Question 1. TDTGP: The components of TDTGP are not described in detail. Please describe in detail the types of constituent polysaccharides and saponins contained in TDTGP.

Answer: More details have been added in Page 2, Lines 67-78 in the revised manuscript

TDTGP is composed of polysaccharides of Asparagus cochinchinensis (Lour.) Merr. (84.08% total polysaccharide purity, 3.1% moisture), total saponins of Panax notoginseng (Burkill) F.H. Chen ex C.H. (mainly composed of 7.20% notoginsenoside R1, 29.39% ginsenoside Rg1, 4.29% ginsenoside Re, 28.38% ginsenoside Rb1, and 6.77% ginsenoside RD for a total of 76.03%) were mixed with silica (Zhengzhou Yuanze Chemical Co., Zhengzhou, China) in a ratio of 5: 1: 2. The polysaccharides of Asparagus cochinchinensis (Lour.) Merr. and the total saponins of Panax notoginseng (Burkill) F. H. Chen ex C. H. were extracted in the same way as in the previous study (16).

Finally, the total polysaccharide content was 56.90%, the total saponin content was 8.93% (0.901% notoginsenoside R1, 3.413% ginsenoside Rg1, 0.481% ginsenoside Re, 3.354% ginsenoside Rb1, and 0.779% ginsenoside RD, respectively), and the content of silica and other substances was 31.17% per 1 g TDTGP.

Question 2. Group name: Group names are not unified in the MS. For example, T4 is a mixture of T4 and T-4.

Answer: Group names have been unified in the revised manuscript.

Question 3. L53-55: This reference evaluated the effects of green tea polyphenols on mice. I think that studies on crustaceans, including shrimp, should be given as references.

Answer:The reference has been replaced with relating to crustaceans in Page 2, Lines 52-54 in the revised manuscript:

. “Dietary Xiao-Chaihu Decoction (14) improved immune function and antioxidant capacity in L. vannamei, and significantly reduced abundances of Vibrio and increased abundances of Ruegeria in the gut.”

Reference:

  1. Su C., Liu X., Lu Y., Pan L., Zhang M. Effect of dietary Xiao-Chaihu-Decoction on growth performance, immune response, detoxification and intestinal microbiota of pacific white shrimp (Litopenaeus vannamei). Fish & shellfish immunology. 2021, 114, 320-329. https://doi.org/10.1016/j.fsi.2021.05.005

Question 4. L69: What is the origin of Silica?

Answer: The silica is supplied by Zhengzhou Yuanze Chemical Co., Zhengzhou, China. The origin of Silica, “silica (Zhengzhou Yuanze Chemical Co., Zhengzhou, China)”has been added in Line 71 of the revised manuscript.

Question 5. L89: Capitalize the first letter of “stress experiment”.

Answer: “2.3. Stress experiment”, the revision has been made in Page 3, Line 97 of the revised manuscript.

Question 6. L107: The salinity is shown as 28%, is this correct? It is 10 times denser than normal seawater.

Answer: It should have been "‰". Page 3, L103: seawater salinity of 28 ± 2.0 ‰; Page 4, L115: salinity 28 ± 2.0 ‰ in the revised manuscript.

Question 7. L113: Did the authors not measure the size of the shrimp after the experiment period? I think this is very important data.

Answer: Length and weight data for shrimp were measured after the 35-day feeding trial, and as the results of these data were similar to those of the previous study, they are not presented in this paper; moreover, this paper is primarily concerned with changes in the transcriptome and gut microbiota. At your suggestion, we have included the length and weight data in the supplementary material for those who wish to check them.

Supplementary materials:

Fig. S1. The body index changes of the L. vannamei in each group of the feeding experiment. Note: C: blank control group; T2 to T8: TDTGP-2 group to TDTGP-4 group. The parameters were calculated as follows: Weight gain (WG; g) = Wt – W0; Percent weight gain (PWG; %) =100 × (Wt – W0)/W0; Body length growth (BLG; cm) = Lt –L0; Percent body length growth (PBLG; %) =100 × (Lt –L0)/L0; Fatness degree (FD) = 100 × Wt/Lt3; Where Wt is the final body weight (g), W0 is the initial body weight (g), Lt is the final body length (cm), L0 is the initial body length, and t is the experimental duration in days.

Question 8. Section 2.9: It is not indicated which post hoc test was used. Please show post hoc test in this section.

Answer: The Duncan-t-test was used for pairwise comparisons among groups. It has been added to Page 5, Line 170 in the revised manuscript.

The SPSS 22.0 software was used to analyze the data. At P < 0.05 level, the main effect was tested by means of one-way ANOVA. The Duncan-t-test was used for pair-wise comparisons among groups.

Question 9. L173-174: Figure 1B does not contain the Flavobacterium. In the text, is Flavobacterium a mistake for the Flavobacteriaceae.

Answer: Flavobacterium has been changed into Flavobacteriaceae in Page 5, Line 182 of the revised manuscript.

Analysis of the abundance of microbes in the intestine of the TDTGP-4 group and the blank control group showed that the abundance of Vibrionaceae in the intestine of the TDTGP-4 group decreased relatively, and the abundance of Flavobacteriaceae in the in-testine of the TDTGP-4 group increased (Fig. 1B).

Question 10. Figure 2 and 6: Despite the fact that there are not very large differences, there are significant differences among the groups. An explanation of which test method was used is needed (same as comment 8). Also, is this minute change a physiologically significant change?

Answer: Both Figure 2 and Figure 6 showed the Duncan multiple comparison method, which is explained in detail in Comment 8.

As L. vannamei is mainly non-specific immune, PO, ACP, i-NOS, SOD and T-AOC can to some extent indicate the immune function and antioxidant capacity of L. vannamei; although only small changes in these indicators are shown in Figure 2 and Figure 6, we believe that this may represent a trend indicating that TDTGP can regulate the immune function of L. vannamei. For example, in a study of Xiao-Chaihu Decoction (1), which regulates the immune capacity and antioxidant function of L. vannamei, feeding different doses of Xiao-Chaihu Decoction also showed only minor changes in hemolymph SOD and T-AOC. Their findings are similar to ours.

Reference:

  1. Su C., Liu X., Lu Y., Pan L., Zhang M. Effect of dietary Xiao-Chaihu-Decoction on growth performance, immune response, detoxification and intestinal microbiota of pacific white shrimp (Litopenaeus vannamei). Fish & shellfish immunology.2021,114,320-329. https://doi.org/10.1016/j.fsi.2021.05.005

Question 11. Figure 3A: It states that the PCA results showed significant differences between the groups (in L201-203). Is it possible to tell from this figure alone whether there is a significant difference? L278-282 and L312-313 are described in the same way, but without evidence.

Answer: These expressions have been modified and the PCA or PcoA plots alone do not really prove significant differences between the samples, only that their genotypes and colony composition have changed in Page 6, L212-214 of the revised manuscript.

PCA showed that, between the TDTGP-4 + ammonia stress group and TDTGP-4, there was a difference in gene expression (Fig. 3A) in Page 8, Lines 250-251 of the revised manuscript.

There was a difference between the ammonia stress group and the other three groups, according to Principal Coordinate Analysis (PCoA) (Fig. 5A) in Page 10, Lines 292-294 of the revised manuscript.

PCA showed that there was a difference in gene expression composition between nitrite stress group, TDTGP-4 group, and blank control group, but the DGEs composition of TDTGP-4 + nitrite stress group was similar to that of TDTGP-4 group (Fig. 7A) in Page 13, Lines 325-326 of of the revised manuscript.

PCoA showed that there was a difference between the nitrite stress group and the blank control group in gut microbial composition.

Question 12. L338-339: Is this content true? On the contrary, there are some genes that are fluctuating. More careful explanation is needed.

Answer: The Lines 338-339 results showed some variation, but only for individual genes between the two groups, which could be due to two reasons: (1)firstly, the large differences in gene expression between individual L. vannamei;(2) secondly, the different calculation methods of RNA-seq and qPCR. qPCR only detects a segment of pu and VEGF of about 200 bp, but RNA-seq is a sequencing analysis of the entire extracted gene, and the data accuracy of RNA-seq is relatively higher than that of qPCR. In this test, the trend agreement between RNA-seq and qPCR was 94.28% (33/35◊100%), which we believe is sufficient to demonstrate the reliability of RNA-seq.

   The changes has been added in Page 13, Lines 351-355 in the revised manuscript:

The results showed that the qPCR results were generally consistent with the trends of the sequencing results, with differences in individual genes in only a few groups (VEGF in the NT4 group and Pu in the T4 group), probably due to large differences in gene expression between individual L. vannamei, but the consistency of the trends in the other data was sufficient to prove that the sequencing results were reliable.

Question 13. Discussion: The relationship between gut microbiota and antioxidant capacity and immune function has not been discussed. Please discuss the relationship between gut microbiota and antioxidant capacity and immune function.

Answer: The relationship between gut microbiota and antioxidant capacity and immune function was discussed in Page 17, Lines 488-499 of the revised manuscript.

“Several studies have shown that gut microbiota are closely related to host immune function and antioxidant capacity; for example, gut microbiota enhance host antioxidant capacity through the generation of reactive sulfur species (30). This suggests that gut microbiome and their metabolites may contribute to host antioxidant capacity and immune function. It has been shown that pumpkin juice fermented by Rhodobacter sphaeroides can improve the antioxidant capacity of pumpkin juice in vitro and in-crease the stability of the gut microbiome in mice (39). Carotenoids with antioxidant activity can be produced by Flavobacteriaceae (40), suggesting that the by-products of both bacteria may secrete substances with antioxidant activity that may modulate host immune functions to some extent. In this study, there was an increase in the abundance of Rhodotoraceae and Flavobacteriaceae in the gut and some increase in immune function after TDTGP treatment, but the direct relationship between the two needs to be further investigated and established.”

Question 14. Discussion: Please discuss that the relationship between the components of TDTGP and antioxidant capacity and immune function in L. vannami.

Answer: The discussion of the relationship between the components of TDTGP and the antioxidant capacity and immune function of L. vannami has been added in Page 17, Lines 387-402 of the revised manuscript:

Plant extracts can modulate the immune and antioxidant function of L. vannamei (22), in this experiment, immune and antioxidant indices such as PO, ACP, i-NOS, SOD, and T-AOC in the hemolymph of the ammonia and nitrite stress groups showed different degrees of increase after ammonia and nitrite stress, indicating that moderate environmental stress can improve the immune function of L. vannamei. Furthermore, we observed that the TDTGP group had higher immune parameters such as PO, ACP, i-NOS, SOD, and T-AOC than in the ammonia and nitrite stress groups after ammonia stress and nitrite stress,which was similar to previous studies and again confirmed that TDTGP could improve the immunity of L. vannamei to some extent (15). This may be related to the composition of TDTGP. Some studies have reported that aqueous root extract of Asparagus cochinchinensis (Lour.) Merr. can increase SOD, CAT and i-NOS activities in the blood of mice, thus improving their antioxidant capacity (23). Panax notoginseng extract (PNE) (24) can increase SOD activity and T-AOC content in the liver of hybrid grouper. These studies have shown that both aqueous root extract of Asparagus cochinchinensis (Lour.) Merr. and Panax notoginseng extract have better immune-enhancing and antioxidant effects, which was consistent with our experimental results.

Question 15. L433-435: Ackermania and Arabidopsis thaliana are plants. Please review the text.

Answer: We double-checked the original reference, which is Akkermansia muciniphila and Parabacteroides distasonis, and have corrected it in Lines 455-L457 of the revised manuscript.

The gut microbiome is moderated by plant extracts. For example, treatment with Panax notoginseng saponins (PNS) shaped the murine gut microbiome by increasing the abundances of Akkermansia muciniphila and Parabacteroides distasonis (35).

Round 2

Reviewer 3 Report

I am satisfied with the revisions that have been made by the authors.